evolution, genetics

Lake Victoria, cichlids, sympatric speciation, genetic architecture, male coloration

**Author for correspondence:**
Anna F. Feller
e-mail: anna.feller@eawag.ch

# Genetic architecture of a key reproductive isolation trait differs between sympatric and non-sympatric sister species of Lake Victoria cichlids

Anna F. Feller[1,3], Marcel P. Haesler[1,3], Catherine L. Peichel[2] and Ole Seehausen[1,3]

[1]Division of Aquatic Ecology and Evolution, Institute of Ecology and Evolution, and [2]Division of Evolutionary Ecology, Institute of Ecology and Evolution, University of Bern, 3012 Bern, Switzerland
[3]Department of Fish Ecology and Evolution, Centre of Ecology, Evolution and Biogeochemistry, EAWAG Swiss Federal Institute of Aquatic Science and Technology, 6047 Kastanienbaum, Switzerland

(iD) AFF, 0000-0001-5786-7658; CLP, 0000-0002-7731-8944; OS, 0000-0001-6598-1434

One hallmark of the East African cichlid radiations is the rapid evolution of reproductive isolation that is robust to full sympatry of many closely related species. Theory predicts that species persistence and speciation in sympatry with gene flow are facilitated if loci of large effect or physical linkage (or pleiotropy) underlie traits involved in reproductive isolation. Here, we investigate the genetic architecture of a key trait involved in behavioural isolation, male nuptial coloration, by crossing two sister species pairs of Lake Victoria cichlids of the genus *Pundamilia* and mapping nuptial coloration in the F2 hybrids. One is a young sympatric species pair, representative of an axis of colour motif differentiation, red-dorsum versus blue, that is highly recurrent in closely related sympatric species. The other is a species pair representative of colour motifs, red-chest versus blue, that are common in allopatric but uncommon in sympatric closely related species. We find significant quantitative trait loci (QTLs) with moderate to large effects (some overlapping) for red and yellow in the sympatric red-dorsum × blue cross, whereas we find no significant QTLs in the non-sympatric red-chest × blue cross. These findings are consistent with theory predicting that large effect loci or linkage/pleiotropy underlying mating trait differentiation could facilitate speciation and species persistence with gene flow in sympatry.

## 1. Background

The adaptive radiation of Lake Victoria haplochromine cichlids comprises approximately 500 endemic species that have evolved within the lake in perhaps as little as 15 000 years [1–4] and that are highly diverse in morphology, ecology, colour and behaviour [5–7]. Typically, however, closely related species are similar in morphology and ecology while they differ dramatically in male nuptial coloration [6,8,9]. Male nuptial coloration is considered a trait of key importance in the origin and maintenance of new species in these fish [9–11].

A highly recurrent pattern in male nuptial colour variation in pairs of closely related species of Lake Victoria cichlids is that males in one species are blue-grey on their body with any red colour confined to the fins, whereas males of the other species are yellow-red on the body [9–12]. The red colour can be confined to either dorsal parts of the body ('red-dorsum' type) or to the chest and lower head ('red-chest' type). When closely related species are sympatric, the red form generally has a red-dorsum, whereas many non-sympatric pairs of closely related species involve a red-chest and a blue form [6,8,13].

A representative case of sympatric red-dorsum and blue sister species is the young species pair of *Pundamilia* sp. 'nyererei-like' and *Pundamilia* sp.

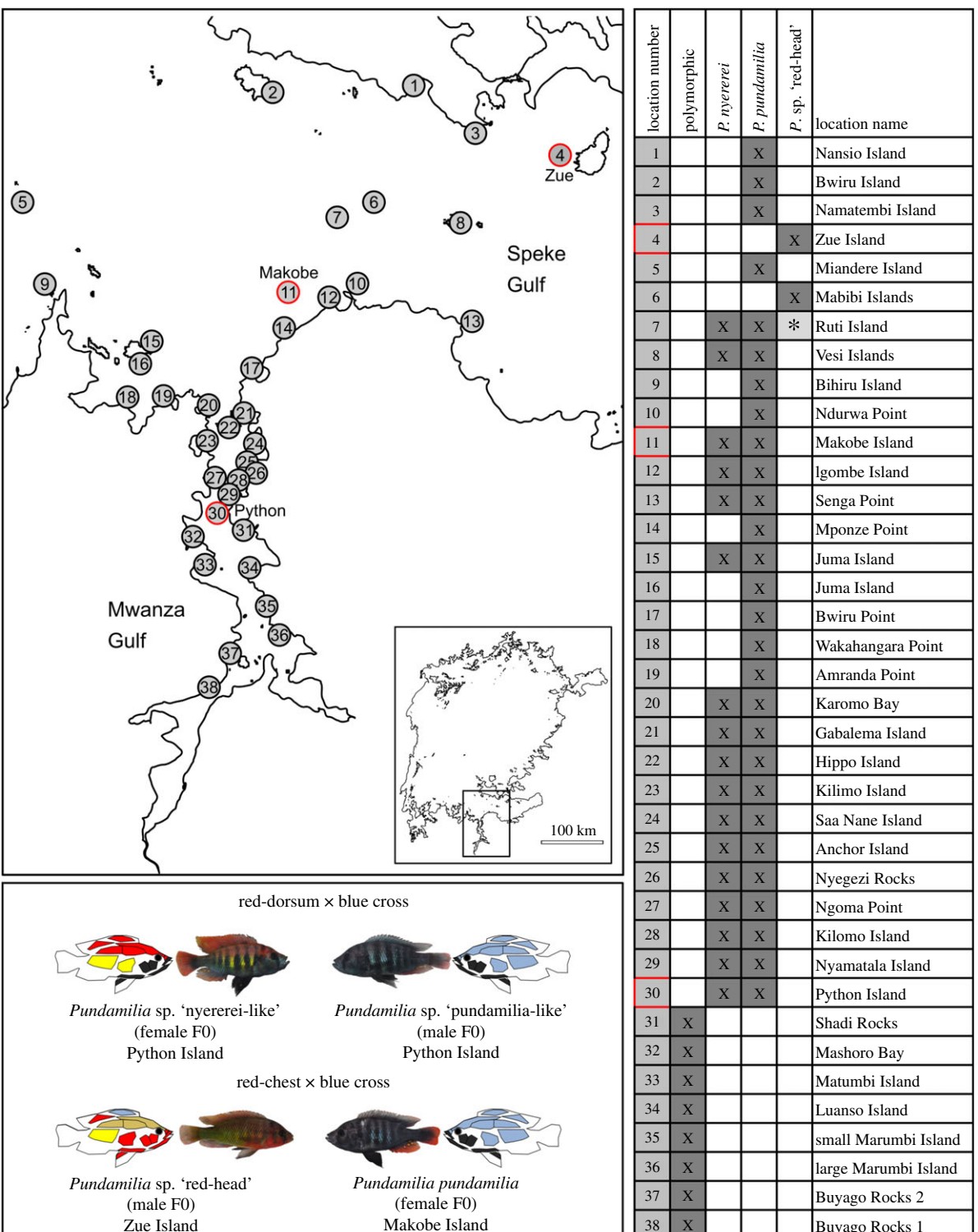

**Figure 1.** Distribution and sampling locations of blue, red-dorsum and red-chest *Pundamilias*. The upper left panel shows the Mwanza gulf region in Lake Victoria (the inset figure indicates the location of this region on the full Lake Victoria map) with all sampling locations (circles with numbers). The table on the right indicates which species were recorded at each sampling location (data assembled from [6,8,13,14]). *P. pundamilia* records include records of *P.* sp. 'pundamilia-like', and *P. nyererei* records include records of *P.* sp. 'nyererei-like' [15,16]. At Ruti island (*), one single *P.* sp. 'red-head' male was caught by O.S. in 2001, but no other individuals of this species were seen in many hundreds of sampled individuals at this location in 10 years before or after this. Sampling locations 31–38 harbour colour-polymorphic *Pundamilia* populations, where species are not clearly identifiable. Highlighted in red are the catch locations of the four parental species of the two crosses, see lower left panel showing representative male individuals of these four species and their colour phenotypes. Lake Victoria contours were taken from the Lake Victoria Bathymetric map V6 by [17] and smoothed in Inkscape™ 0.92.

'pundamilia-like' at Python Island (figure 1). This pair may have evolved in sympatry from a hybrid population in only a few hundred generations [15,16,18]. *Pundamilia* sp. 'red-head' and *Pundamilia pundamilia* are representatives of a red-chest and blue species with overlapping geographical distributions that are never found together on the same rocky island, despite islands occupied by the different species lying within dispersal distance of each other (figure 1) [6,8,19].

Male nuptial colour in *Pundamilia* is under intra- and intersexual selection. Males fiercely compete for territories, and they use their bright colours to signal territory ownership to contestants. Negative frequency-dependent selection

generated by own-type aggression biases is likely involved in stabilizing the coexistence of different colour types [11,20–22]. Females, which are cryptically coloured and the sole investors into parental care, exhibit strong preferences for mates with particular nuptial coloration, which generates both directional sexual selection within and assortative mating between species [23–26]. Preference for male colour has been shown to be heritable and likely determined by few major genes or genomic regions [27,28] and to generate disruptive selection on male colour [29].

Despite strong preferences for bright male colour and assortative mating, there is some gene flow in fully sympatric red-dorsum versus blue pairs such as Pundamilia sp. 'nyererei-like' and Pundamilia sp. 'pundamilia-like' [15,16]. Nonetheless these two species persist in sympatry. With a genome-wide mean FST of 0.053 and hundreds of highly differentiated genomic regions, they show surprisingly strong genetic differentiation considering they have likely evolved in full sympatry in less than 200 generations [15,16]. Red-chest versus blue pairs like Pundamilia sp. 'red-head' and Pundamilia pundamilia experience little gene flow because they usually do not co-occur. Although they are neither strictly geographically isolated nor dispersal limited [6,8,19] (and figure 1), they do not seem to persist as two species in the same island.

Here we ask whether differences in the genetic architecture of the male nuptial colour motifs that differentiate these species could explain the difference in the distribution patterns of the species pairs. We tested the hypothesis that red-dorsum versus blue pairs persisting in sympatry despite some gene flow have an architecture that is robust against potentially homogenizing effects of gene flow, while the absence of such an architecture would make it difficult for red-chest versus blue pairs to persist in the presence of gene flow.

In a sympatric scenario with ongoing gene flow, recombination is expected to break up linkage disequilibrium between favourable combinations of alleles for local adaptation and reproductive isolation [30,31]. This is less likely to occur if divergently selected traits are coded by few genes with large effects, as this reduces the number of targets for recombination to break up, and increases the effectiveness of (correlational) selection because it is concentrated on fewer targets [32]. Indeed, theoretical work has shown that large effect alleles, or groups of tightly linked alleles with smaller effects that then act like a large effect locus, are less likely to be lost when adaptive multilocus phenotypes need to be maintained against gene flow [33–35]. Similarly, the physical linkage of multiple traits (or pleiotropy) could facilitate divergence with gene flow [34,36,37].

Recombinant males between cichlid (Pundamilia) species with either red or blue male nuptial coloration might have reduced fitness if they will be less likely to be chosen by females with strong preferences for either colour, as is the case in these species [23–28], or if intermediate coloration makes them targets of territorial aggression by males of both species. We thus predict large effect loci and/or physical linkage of several loci (or pleiotropy) for male nuptial colour to be present in sympatric red-dorsum versus blue cichlid pairs, as such an architecture could both facilitate the establishment of polymorphisms [38] and make it easier to retain phenotypic differentiation in sympatry with ongoing gene flow (see above). Furthermore, theoretical models that investigate the feasibility of sympatric speciation by sexual selection (alone) usually find such speciation feasible when

assuming a simple genetic architecture (i.e. few additive loci with large effects) and that speciation becomes less likely when the number of loci underpinning reproductive isolation traits increases [39–41].

To compare the genetic architecture of red and yellow versus blue male nuptial colour motifs that do or do not persist with gene flow, we crossed Pundamilia sp. 'nyererei-like' and Pundamilia sp. 'pundamilia-like' (red-dorsum × blue) and Pundamilia sp. 'red-head' and Pundamilia pundamilia (red-chest × blue) in the laboratory and performed quantitative trait locus (QTL) mapping analyses on male nuptial coloration in the second generation (F2) hybrids.

We find several significant QTLs for the presence/absence of red and yellow colour in the sympatric red-dorsum × blue cross, with several traits mapping to the same region, whereas we find no significant QTLs in the non-sympatric red-chest × blue cross. We conclude that the presence of large effect loci, with physical linkage between some traits (or pleiotropy), likely makes up one key element for the rapid evolution of reproductive isolation and species persistence in sympatry despite some gene flow.

## 2. Material and methods

### (a) Experimental crosses

The red-dorsum × blue cross was started with a Pundamilia sp. 'nyererei-like' female and a Pundamilia sp. 'pundamilia-like' male, both from laboratory bred strains established from fishes caught by OS at Python Island in Lake Victoria in 2003 (figure 1). The red-chest × blue cross was started with a Pundamilia pundamilia female caught by O.S. at Makobe Island in 2003 and a Pundamilia sp. 'red-head' male from a laboratory bred strain established from fishes caught by O.S. at Zue Island in 1993 (figure 1). Five to 6 days after spawning, the eggs were removed from the female's mouth and reared in an egg tumbler until hatching. After yolk sac resorption, the larvae were transferred to rearing aquaria that were part of a large recirculation system. After sexual maturity at the age of 1 to 2 years, two pairs of F1 individuals of each cross were then allowed to mate, and the eggs and larvae were reared the same way as the F1 generation. Because the average clutch size is just about 20–30 juveniles, we re-mated each F1 pair multiple times until we had obtained a total of approximately 300 F2 individuals in each cross. This procedure of repeatedly re-mating the same pairs took about 2 years. All F2s were reared to an age of at least 1 year before they were phenotyped. All four populations from which the grandparents were taken breed true in the laboratory in a common garden environment, hence the differences in coloration are heritable in both crosses.

### (b) Colour photos and scoring of coloration

Sexually mature F2 males were removed from their holding tank and individually placed into one of six adjacent plexiglas photo cuvette compartments with transparent separations and a grey PVC background, inside a larger aquarium. This set-up induced territoriality in the males who could see each other and hence make them express full colour. Two Walimex pro Daylight 250S lamps were placed on either side of the cuvette and a first colour picture was taken of each fish with a Canon D60 camera equipped with a 50 mm lens (settings; P mode, ISO 200, auto focus). If a male failed to show territorial behaviour within an adjustment time of 1–2 days, it was moved back into its home tank and the process was repeated several weeks later. After good photos were obtained, each fish was sedated in MS222

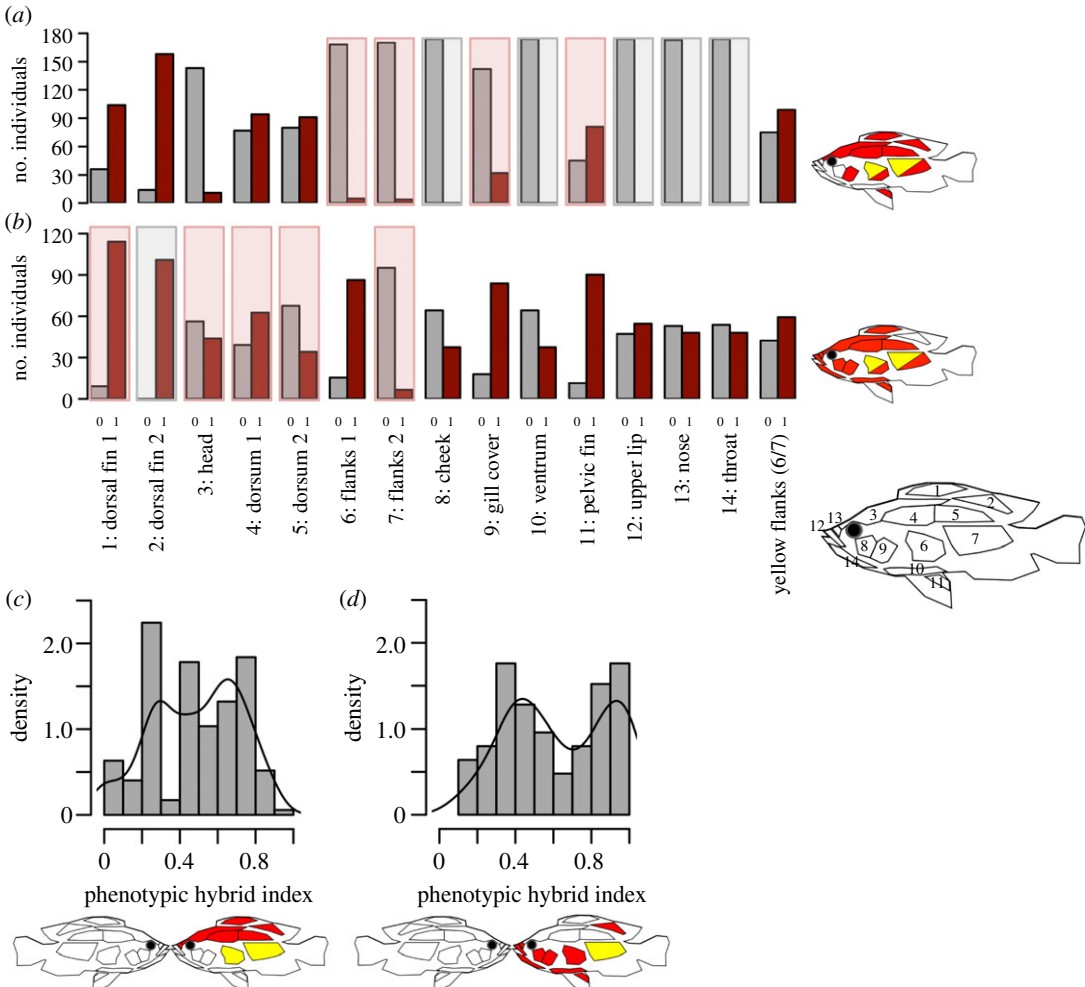

**Figure 2.** Phenotypic distributions in the two crosses. Distribution of absence (0; grey)/presence (1; red) scores for all traits in the (*a*) red-dorsum × blue cross and (*b*) red-chest × blue cross. A grey background indicates lack of variation in the F2 individuals in this trait (i.e. non-mappable traits), a red background indicates traits that are not differentiated between the two parental species but were variable in the F2 hybrids (transgressive traits). The sectors highlighted with colour on the inset cartoon fish are the traits that were mappable in the cross. Half red/half yellow indicates either colour could occur in this trait. (*c*,*d*) The distribution of the standardized phenotypic hybrid index, for the red-dorsum × blue cross (*c*) and for the red-chest × blue cross (*d*), which was calculated for every individual as the sum of presence scores in all traits differentiating the two parental species divided by the number of differentiating traits (see parts highlighted with colour in the inset fish figures). An index of 0 hence corresponds to the phenotype of the blue species, an index of 1 corresponds to the phenotype of the red species. Note that no red was scored on the nose on any F2 individual in the red-dorsum × blue cross. The comparison of the cartoon fish indicating mappable traits in (*a*,*b*) with the cartoon fish showing the parental types in (*c*,*d*) highlights which traits are transgressive in the F2 hybrids.

(50 mg l$^{-1}$), euthanized in MS222 (300 mg l$^{-1}$) and immediately put on melting ice for 10 min. This treatment had the effect of relaxation of the melanophores of system I that are under nervous control [42] and made colour better visible. The fish were then pinned down in a dissection bowl containing a wax layer and a ruler with a colour reference, and submerged in water (just covering the fish), and a second colour picture was taken from above with the same camera (same settings), now equipped with an 18–135 mm objective and with one Walimex pro Daylight 250S lamp for lighting. We applied a white balance filter to these second photos in Adobe Photoshop CC 2018 by setting a white point on the white part of the colour bar ruler.

We visually scored the presence/absence of red on 14 sectors of the body (henceforth referred to as traits) that are different between at least two of the parental species (see figures 1 and 2), and the presence/absence of yellow flanks (as 15th trait) from the second photo (when the fish was dead). The first pictures were only used as a cross-reference to confirm uncertain scores. In some individuals (mostly of the red-dorsum × blue cross), some traits could not be scored unambiguously and were scored as NA. In addition, we calculated a phenotypic hybrid index for each individual as the summed up presence scores (NAs treated as 0.5) in the subset of traits that differentiate the two respective parental species divided by the number of

traits in the respective subset (figure 2*c*,*d*). The phenotypic hybrid index thus ranges from 0 (all traits like those of the blue species) to 1 (all traits like those of the red species).

### (c) DNA extraction and RAD sequencing

Genomic DNA was extracted from finclips, which had been stored in 98% ethanol, using phenol–chloroform [43]. We prepared restriction-site associated DNA (RAD) sequencing libraries using *SbfI* as restriction enzyme, following [44] with some modifications (see electronic supplementary material, appendix S1). Single end-sequencing (100 bp for all others but 125 bp for the last two libraries, see electronic supplementary material, table S5) was done on an Illumina HiSeq 2500 platform either at the Genomic Technologies Facility of the University of Lausanne or at the Next Generation Sequencing Platform of the University of Bern. Each library was sequenced on a single lane. Bacteriophage PhiX genomic DNA was added to each library (4–12% of reads) to increase complexity in the first 10 sequenced base pairs and for base quality recalibration (see below).

### (d) Sequence processing and genotyping

Demultiplexed, trimmed and filtered reads (see appendix S2 for details) were aligned to the anchored version of the *Pundamilia*

nyererei reference genome [45] with Bowtie2 v2.3.2 [46], allowing one mismatch. This was followed by base quality recalibration (see electronic supplementary material, appendix S2 for details) and subsetting to uniquely aligned reads. GATK Unified Genotyper [47] was used for genotyping (minimum base quality score set to 20). The resulting vcf files were filtered with Bcftools implemented in Samtools v.1.8 [48], Vcftools v.0.1.14 [49] and a custom Python script, to obtain bi-allelic SNPs (see electronic supplementary material, appendix S2 for details). In the red-dorsum × blue cross, 368 SNPs were homozygous fixed in the F0 and heterozygous in all four F1 parents, and were used in linkage map construction. In the red-chest × blue cross, 2358 SNPs were homozygous fixed in the F0 and heterozygous in two F1 parents (the other two F1s had low quality data that was removed during filtering), and were used in linkage map construction.

## (e) Linkage map construction

We used JoinMap 4.0 [50] to build linkage maps for both crosses. We removed loci with extreme segregation distortion ($p < 0.01$), loci with greater than 20% missing genotypes and identical loci (i.e. SNPs within the same RAD locus) (greater than 0.950). Individuals were removed if they had greater than 30% missing data. The linkage maps were generated from 216 F2 individuals (173 males, 43 females) in the red-dorsum × blue cross and 171 F2 individuals (115 males, 56 females) in the red-chest × blue cross. We identified linkage groups based on an independent logarithm of odds (LOD) threshold of 5. Loci with suspicious linkage (recombination frequency greater than 0.6) were removed. The strongest cross-link (SCL) values in the maps are 4.7 (red-dorsum × blue cross) and 4.6 (red-chest × blue cross), and unlinked markers were excluded. To build the linkage maps, we used the Kosambi regression mapping algorithm with a LOD threshold of 1.0, a recombination threshold of 0.499, a goodness-of-fit threshold of 5.0 and no fixed order. We performed two rounds of mapping with a ripple after addition of each marker to the map (see [50]).

## (f) QTL mapping

QTL mapping of male nuptial colour traits was performed in R/qtl [51,52]. We mapped the presence/absence of yellow colour on the flanks and of red colour in 14 body and fin locations (traits; see above and figure 2) in F2 males ($n = 174$ in the red-dorsum × blue cross and $n = 125$ in the red-chest × blue cross) as binary traits. Conditional genotype probabilities were calculated using the calc.genoprob function with a fixed stepsize of 1 (respectively 3 for scantwo; in cM), an assumed genotyping error rate of 0.05, and the Kosambi map function. The scanone and scantwo functions were used with the EM algorithm, and significance thresholds were determined by permutations ($n = 1000$). For the red and yellow traits, we used the binary model; for a multi-trait hybrid index (see below), we used the normal model. We consider $p < 0.05$ as significant, and $p < 0.1$ as marginally significant. For traits showing a significant effect of F1 family, mapping was additionally performed with family as an additive covariate (allowing the average phenotype in the two families to be different) and as both an additive and interactive covariate (additionally allowing the effect of the QTL between the two families to be different). For significant QTLs, approximate Bayesian credible intervals with a 0.95 probability coverage were calculated using the bayesint function, and the percentage of variation explained (PVE) of significant QTLs was calculated for each trait individually using the fitql function (with the HK algorithm since the EM algorithm was not available in this function for binary models).

The lower number of individuals in the red-chest × blue cross could result in reduced power of detecting QTLs of similar effect sizes as compared to the red-dorsum × blue cross, and the two linkage maps differ substantially in number and density of markers. To account for this, we repeated the single QTL analyses for the red-

dorsum × blue cross after randomly downsampling to 125 (of 174) individuals using the sample function in R to randomly pick individuals and then subsetting the genotype–phenotype file to these individuals. For the red-chest × blue cross, we repeated the single QTL analyses after randomly downsampling markers on the linkage map to match the number of markers on each linkage group to those in our sparser red-dorsum × blue map, again using the sample function in R to randomly pick markers within each linkage group and then subsetting them to these markers. The procedure was repeated five times in each cross.

# 3. Results

## (a) Linkage maps

The final map for the red-dorsum × blue cross contains 232 markers in 22 linkage groups (corresponding to the number of expected chromosomes in haplo-tilapiine cichlids [53,54]) with an average marker distance of 5.3 cM and a total map length of 1117.8 cM. The final map for the red-chest × blue cross contains 1198 markers in 22 linkage groups with an average marker distance of 1.2 cM and a total map length of 1360.8 cM.

## (b) Phenotypic distributions

Seven traits differentiate the parental species in the red-dorsum × blue cross (figure 2c): both parts of the dorsal fin, both parts of the dorsum, head and nose are red versus blue, and both parts of the flank (considered as one trait) are yellow versus blue. However, nose was never scored as red in any F2. Red was also scored in some F2s for both parts of the flank, for gill cover, and for pelvic fin, although neither parental species has red in these traits (figures 1 and 2c). Altogether this resulted in 10 mappable traits for this cross (figure 2a). The F2 phenotypes range from no red or yellow, respectively, in any of the traits that differentiate the parental species (i.e. like *P.* sp. 'pundamilia-like'), to red or yellow, respectively, in all of these traits with the exception of nose (i.e. like *P.* sp. 'nyererei-like'), with most individuals being intermediate in expressing red in some but not all of these traits (figure 2c).

Ten traits differentiate the parental species in the red-chest × blue cross (figure 2d): the rear dorsal fin part, the frontal flank part, the pelvic fin and all parts on the head (except for the head part itself, which like the dorsum is greenish versus blue) are red or yellow (flanks) versus blue. The rear dorsal fin part was scored as red in all F2s and can hence not be mapped. Some F2s were scored as red on the front part of the dorsal fin as well as on both parts of the dorsum, on the head and on the rear flank part, even though neither parental species is red there (figures 1 and 2d). Altogether this resulted in 14 mappable traits for this cross (figure 2b). The F2 phenotypes range from no red or yellow, respectively, in any of these traits with the exception of the rear dorsal fin part (i.e. like *P. pundamilia*), to red or yellow, respectively, in all these traits (i.e. like *P.* sp. 'red-head'), with most individuals being intermediate in expressing red in some but not all of these traits. Yet, a large number of these F2 hybrids resemble *P.* sp. 'red-head' whereas fewer resemble *P. pundamilia*, suggesting more directional dominance effects in this cross than in the other cross.

## (c) QTL mapping results

We found significant QTLs for red and yellow colour for seven out of 10 mappable traits (figure 2a) in the sympatric red-

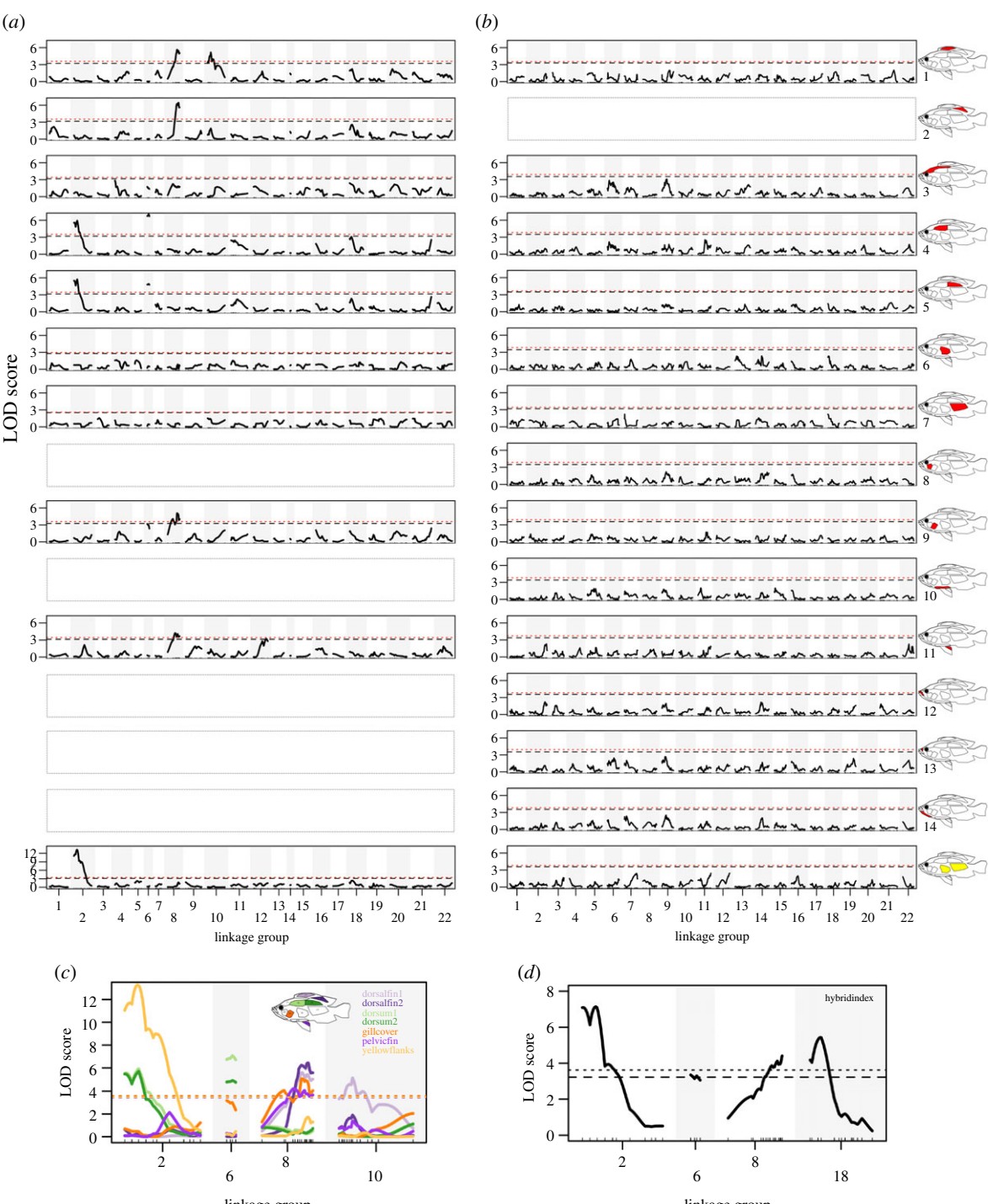

**Figure 3.** QTL mapping of the presence/absence of red (and yellow) male nuptial colour. LOD scores across the 22 *Pundamilia* chromosomes for all traits with presence scores for red and for yellow flanks (empty plots for non-mappable traits, figure 2) in (*a*) the red-dorsum × blue cross and (*b*) the red-chest × blue cross. The black dashed lines represent genome-wide significance thresholds of *p* < 0.1 for each trait, the red dotted lines for *p* < 0.05. (*c*) LOD scores across the chromosomes containing significant QTLs for red and yellow in the red-dorsum × blue cross (genome-wide significance thresholds of *p* < 0.05 shown for each trait); (*d*) shows the chromosomes containing significant QTLs for the hybrid index in the red-dorsum × blue cross. See also electronic supplementary material, table S1 and figure S1.

dorsum × blue cross (electronic supplementary material, table S1 and figure S1; figure 3*a,c*). For red in both sectors of the dorsal fin, the gill cover and the pelvic fin, we identified QTLs on Pun-LG8 with overlapping 95% confidence intervals, each with a PVE of 12.6–16.8%. A second QTL for red on dorsal fin sector 1 was found on Pun-LG10 with a PVE of 15.5%, and a second marginally significant QTL for red on the pelvic fin was found on Pun-LG12 with a PVE of 10.8%. Two QTLs for red on both dorsum parts were identified, one on Pun-LG2 and one on Pun-LG6, both with a PVE of

12.4–17.4%. Pun-LG2 also contains a QTL for yellow flanks, within the 95% confidence mapping interval for red-dorsum, with a PVE of 29.5%. In this cross, only red in the pelvic fin (a transgressive trait in this cross) showed a significant effect of family ($F = 26.901$, $p < 0.001$). Repeating the mapping with family as covariate for pelvic fin recovered the QTL on Pun-LG8 and revealed an additional marginally significant QTL on Pun-LG22. We also found four QTLs for the phenotypic hybrid index (figure 3*d*), one each on LG2 (17.2% PVE), LG6 (8.5% PVE), LG8 (10.97% PVE), all of them within the 95%

confidence intervals of the QTLs found for the individual traits and one on LG18 (13.3% PVE).

Two-dimensional two QTL scans revealed several additional putative QTLs in this cross (electronic supplementary material, table S2). Repeating the single QTL mapping with 125 individuals randomly sampled from 174 five times recovered 82% of the expected total of 50 significant QTL results (i.e. 10 significant QTLs in the original dataset times five); 36 were recovered as significant and five as marginally significant (electronic supplementary material, table S3).

We found no significant QTLs for any of the 14 mappable traits (figure 2b) nor for the hybrid index in the non-sympatric red-chest × blue cross (electronic supplementary material, table S1; figure 3b). Several traits showed a significant effect of family in this cross: cheek ($F = 16.478$, $p < 0.001$), ventrum ($F = 29.407$, $p < 0.001$), upper lip ($F = 29.079$, $p < 0.001$), nose ($F = 18.232$, $p = 0.001$), throat ($F = 46.281$, $p < 0.001$) and yellow flanks ($F = 5.6352$, $p = 0.025$). Repeating the mapping with family as covariate for these traits; however, only detected one marginally significant QTL for red on the throat on Pun-LG7 with a PVE of 8.3%. A QTL for red on the head (a transgressive trait in this cross) reached the 0.1 significance threshold when repeating the mapping with a subsampled linkage map in two out of five such mapping rounds (electronic supplementary material, tables S1 and S4). Two-dimensional two QTL scans for this cross revealed two potentially interacting QTLs each for red on the cheek and throat (Pun-LG1 and Pun-LG4 for both traits) and for yellow on the flanks (Pun-LG3 and Pun-LG11) (electronic supplementary material, table S2).

## 4. Discussion

We investigated the genetic architecture of a trait complex of key importance to speciation in Lake Victoria cichlid fish, male nuptial colour motifs that feature importantly in behavioural reproductive isolation. In a cross between two sympatric species, representative in their mating trait motifs of many closely related sympatric species pairs, we found significant QTLs with moderate to large effects for red and yellow colour traits, with several traits mapped to the same genomic regions. These results are consistent with genetic architectures predicted to facilitate differentiation and persistence of differentiation in traits contributing to reproductive isolation in sympatry with ongoing gene flow [32–34,36–38]. By contrast, we did not find any significant QTLs in a cross between two species representative in their mating trait motifs of closely related species that are usually seen to occupy different islands but do not occur in sympatry. This is consistent with our hypothesis of a genetic architecture that makes phenotypic differentiation not robust to gene flow. We argue that these differences in genetic architecture of superficially similar trait differences could help explain why species with the red-dorsum nuptial colour motif are often sympatric with blue sister species, whereas those with the red-chest motif seem unable to retain differentiation from their blue relatives in sympatry.

The difference between the two crosses in the presence/absence of QTLs with moderate to large effects cannot simply be explained by a difference in power to detect QTLs: repeatedly and randomly downsampling the red-dorsum × blue cross F2 individuals to match the lower sample size in our red-chest × blue cross (a lower sample size decreases power to detect QTLs) did not significantly change the results, nor did downsampling the markers on the linkage map in the red-chest × blue cross to match the sparser red-dorsum × blue cross-linkage map (a sparser marker density lowers the significance threshold). Another statistical bias [55], where low sample sizes lead to overestimation of effect sizes or PVE, mainly due to the difficulties of statistically detecting loci with small effects, cannot be ruled out. However, this would be expected to affect both of our crosses and should thus not confound the comparison between the two. The different direction of our two crosses (red female × blue male in the red-dorsum × blue cross, blue female × red male in the red-chest × blue cross) should also not affect our results: the lab-strain populations from which the grandparents were taken have stable male colour. Also, none of the QTLs map to known sex determining chromosomes (Pun-LG10 in the red-chest × blue cross [45]), which is also consistent with earlier studies of experimental crosses of the same red-dorsum versus blue species pair [56–58]. Furthermore, in both crosses, both parental phenotypes (considering traits differentiating the two species, figure 2c,d) are recovered (with the exceptions that one trait (nose) was never scored as red in any F2s in the red-dorsum × blue cross, and one trait (dorsal fin 2) was scored as red in all F2s of the red-chest × blue cross). Additionally, most individuals are intermediate in the number of traits in which red is expressed in both crosses, albeit with signs of more red dominance in the red-chest × blue cross.

A previous crossing experiment [56] estimated that the difference in the amount of red on the body (mostly flank and dorsum) of males between *Pundamilia* sp. 'nyererei-like' and *Pundamilia* sp. 'pundamilia-like' (our red-dorsum × blue cross) is likely controlled by at least 2–4 loci (with effects of dominance and epistasis). Furthermore, they estimated one gene with complete dominance for yellow flank and epistatic interaction with red on flank and dorsum. Our results conform to these estimates quite well: we found two significant QTLs each for red on the dorsum (LG2 and LG6) and for red on the dorsal fin (LG8 and LG10), each with a PVE of 12–17%. The interval of the QTL for red on the dorsum on LG2 also contains a major QTL for yellow on the flanks with a PVE of nearly 30%, and the significant QTLs for pelvic fin and gill cover overlap with the interval of the QTL on LG8 for red on the dorsal fin, suggesting either linkage of several loci or pleiotropic effects of a single locus. Two-dimensional 2-QTL scans indicate the presence of additional loci contributing to red/yellow colour. Although our modest sample sizes do not allow us to detect small effect QTLs, they are likely present, as our QTLs do not explain all of the variance in our mapped traits. However, the main contribution to variance in red and yellow male colour in this cross comes from these four QTL regions. A first screen of the genes closest to the QTL peaks has not yet revealed any obvious candidate genes. To follow-up on screening for candidate genes across the mapping intervals, each of which contains many dozens to hundreds of genes, will be a topic of future work.

The presence of moderate to large effect QTLs should make phenotypic differentiation and maintenance of differentiation in sympatry more likely than a more dispersed architecture of many loci with small effects under the opposing effects of disruptive selection and gene flow e.g. [32–34,38]. Furthermore, the theoretical models of sympatric

speciation by sexual selection suggest such a process is more likely when reproductive isolation is based on traits underlain by fewer loci [39–41]. Additionally, we find that several traits (figure 3c) map to the same chromosomal region. Although our current dataset does not allow us to determine whether this is due to a single pleiotropic locus or due to several tightly linked loci, both pleiotropy and physical linkage favour divergence and persistence of phenotypic differentiation despite gene flow [34,36,37]. Linkage (or pleiotropy) of divergent adaptive traits has also been observed in Midas cichlids, which are undergoing rapid sympatric divergence [59]. Our findings are also similar to other systems such as *Heliconius* [60], *Mimulus* [61] or *Drosophila* [62], in which traits involved in reproductive isolation in the presence of gene flow are underpinned by large effect or pleiotropic loci. However, none of these other studies made a direct comparison of the genetic architecture of corresponding key traits for reproductive isolation in species pairs that persist (and probably evolved) in sympatry and others that do not persist in sympatry, as we have done here.

In our cross between the species that do not persist in sympatry, i.e. *Pundamilia* sp. 'red-head' and *Pundamilia pundamilia* (red-chest × blue), we cannot directly infer the type of genetic architecture underpinning red, or yellow due to the absence of any significant QTLs. Most likely, the effects of potential QTLs are too small to be detected with our sample size, suggesting the presence of a larger number of small effect loci. In red-chest versus blue species without direct geographical contact and with little gene flow that is restricted to rare (but documented, figure 1) long distance dispersal events, trait differentiation due to more and smaller effect mutations is more likely to evolve and persist because recombination will not erode associations between them.

What we cannot yet resolve is whether the genetic architecture for male colour in our red-dorsum versus blue pair has evolved in the face of gene flow, i.e. through selection for the clustering of small effect loci, for instance through genomic rearrangements [63], or was already present, allowing the two species to speciate (and now persist) in sympatry.

## 5. Conclusion

The presence of large effect loci, and of physical linkage or pleiotropy, underlying traits involved in behavioural reproductive isolation, such as male nuptial coloration, may enable sister species pairs to differentiate and persist in sympatry despite some gene flow. One hallmark of the East African cichlid radiations is the rapid evolution of strong (behavioural) reproductive isolation that is robust to full sympatry in many closely related species [64]. If the genetic architecture of male nuptial coloration in sympatric *Pundamilia* species we report here is representative for other Lake Victoria cichlid species that live sympatrically, this may help explain how speciation in this system could have led to the rapid emergence of communities with many closely related species that persist in sympatry, and why some phenotypic motifs regularly distinguish sympatric species while others are confined to allopatric species.

Ethics. Fish experimentation and euthanasia were authorized by the veterinary offices of the cantons of Lucerne (licence number: LU04/07) and Bern (BE 18/15 & BE 65/18).

Data accessibility. Raw read (fastq) files for all genotyped individuals of the red-dorsum × blue cross are deposited in the NCBI short read archive (SRA accession PRJNA612290). The fastq files for the red-chest × blue cross have previously been deposited by [45] (SRA accession PRJNA439430). The filtered vcf files, linkage maps, phenotype–genotype tables, and additional information on sequencing data for both crosses are available from the Dryad Digital Repository at https://dx.doi.org/10.5061/dryad.5tb2rbp1n [65].

Authors' contributions. A.F.F. scored and analysed colour in both crosses, prepared the RAD sequencing libraries for the red-dorsum × blue cross, processed the sequencing data, generated linkage maps, performed the colour QTL analyses in both crosses, wrote the manuscript together with O.S. M.P.H. reared the fish, took the colour photos, prepared all the red-chest × blue and two of the red-dorsum × blue cross RAD sequencing libraries and provided critical feedback for the manuscript. C.L.P. provided guidance for the analyses and contributed to manuscript writing. O.S. designed and coordinated the study, developed the colour scoring scheme and the colour trait space, collected the species distribution data, wrote the manuscript together with A.F.F.

Competing interests. We declare we have no competing interests.

Funding. This work was funded by Swiss National Science Foundation (SNSF) grant nos. 31003A_163338, 31003A_144046 and 31003A_118293 to O.S.

Acknowledgements. We thank Julia Schwarzer and Philine Feulner for their earlier and preliminary work on the red-chest × blue cross and Oliver Selz for initiating the red-dorsum × blue cross. Many thanks to Salome Mwaiko for assistance in the laboratory, the FishEc team for helpful discussions and feedback throughout the process. Joana Meier for providing the allelic balance filter script. The bioinformatics facilities for sequence data processing were provided by the Genetic Diversity Centre at ETH Zürich. We further thank two anonymous reviewers for their constructive feedback on a previous version of this manuscript.

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
