## [Reviewer comments · Proceedings of the Royal Society B: Biological Sciences]

Review History

RSPB-2019-2310.R0 (Original submission)

Review form: Reviewer 1

Recommendation

Major revision is needed (please make suggestions in comments)

Scientific importance: Is the manuscript an original and important contribution to its field?

Good

General interest: Is the paper of sufficient general interest?

Good

Quality of the paper: Is the overall quality of the paper suitable?

Acceptable

Is the length of the paper justified?

Yes

Should the paper be seen by a specialist statistical reviewer?

No

Do you have any concerns about statistical analyses in this paper? If so, please specify them explicitly in your report.

No

It is a condition of publication that authors make their supporting data, code and materials available - either as supplementary material or hosted in an external repository. Please rate, if applicable, the supporting data on the following criteria.

Is it accessible?

Yes

Is it clear?

Yes

Is it adequate?

Yes

Do you have any ethical concerns with this paper?

No

Comments to the Author

The manuscript "Genetic architecture of a key reproductive isolation trait differs between sympatric and non-sympatric sister species of Lake Victoria cichlids" by Feller and colleagues reports on two QTL mapping analyses on two crosses of closely related Lake Victoria cichlid fish.

The manuscript is overall interesting and well-written, and a great fit for Proceedings B so I think that this manuscript will eventually make a great addition to the journal.

For the time being, however, I have a few concerns which prevent me from suggesting acceptance. Addressing these concerns will require changes to the text and most likely partial reanalysis, but they are overall doable and should not undermine the main message of the manuscript. For this reason, I encourage the Authors to put in the necessary effort to improve this manuscript following my comments.

L38: The Authors should better qualify statements about strength of differentiation, particularly to the benefit of a broader audience which may not be familiar with the study system and/or typical divergence in sympatric settings.

Here, the Authors define the differentiation between "nyerei-like" and "pundamilia-like" as "surprisingly strong". However, figure 1b in Meier, et al. (2018) (which the Authors cite as source) reports for this species pair an F_{ST} of 0.053, which most people would find neither strong nor surprising.

I suggest that the Authors provide some information about how strong (F_{ST} or some other relevant results from the study they cite) and, possibly, tone down this statement.

A possibility is - in addition to providing some information about the strength of differentiation - also explaining to the reader why a certain level of differentiation is surprisingly strong in this system.

L49-51: This is correct, but another factor which could contribute to divergence in the presence of gene flow is (loosely defined) linkage of multiple traits, as the organization of genes in genomes will itself create non-random combinations of alleles across multiple loci, and this clearly will be even more fitting in the case of multiple traits being subject to selection and mapping to the same region.

While it is somewhat debatable to which level the different traits mapped in this study represent distinct traits, I think that the explanation above should be considered and explicitly

acknowledged in the Introduction and Discussion. This is because of the substantial level of overlap in QTL regions the Authors find in their study (e.g., Figure 3c).

Relevant work which could be included in such discussion include Fruciano, et al. (2016) for an empirical case in a sympatric system.

Flaxman, et al. (2014) could also be cited for simulation/theory on the general idea that linkage/organization in genomes will favour divergence (which is reflected in the next sentence at L55). The reason why I think Flaxman et al's work is relevant here is that, while the Authors of this manuscript focus on maintenance of phenotypic variation in the face of gene flow (L55-57), the very young age of this system makes it possible that we are now observing a "snapshot" of divergence between two forming species which in the future would go through a quick transition "congealing" phase as per Flaxman et al's simulations, possibly aided by linkage of multiple traits.

Sequence processing, genotyping and map construction - I find that the use of different criteria for the two crosses is neither adequately documented nor justified.

For instance, at lines 154-155, the Authors simply state that the procedure for the red-chest x blue cross was very similar to the one for the other cross, but with a different software for genotyping. However, checking the original paper by Feulner, et al. (2018) the procedure appears substantially different compared to the one in Appendix S2. For instance, Feulner, et al. (2018) use an initial depth threshold of 3 (which is actually quite low) while in Appendix S2 genotypes with a depth <10 are set to missing.

This creates a situation in which the level of reliability of the two datasets/crosses is different (it is hard to evaluate by how much, as the sequential procedures used are different). Then, the radically different number of markers in the maps for the two crosses (368 vs 2052) are probably not only a function of the different level of divergence between parental species, but also probably the inclusion of lower-quality SNPs in the red-chest x blue cross.

The same applies to map construction. While map construction procedure is similar between the two studies, the thresholds and criteria used here are different from the ones in Feulner, et al. (2018).

All of this creates a substantial risk that differences in results between the two crosses are due to difference in statistical power, rather than to real biological differences. I do appreciate and commend the Authors for the downsampling solution they have included (L188-195). However, this solution still does not account for differences in quality/processing between crosses.

First of all, sufficient information on the red-chest x blue cross should be provided in this manuscript/supplementary to avoid that the reader has to check a different paper.

Second, my strongly preferred suggestion is that the analyses are repeated using the same criteria/pipeline for the red-chest x blue cross to ensure a fair comparison between the two crosses.

As a much less preferred option, the Authors should to the very least be very clear to the reader about what the differences between the two crosses are, and what are the consequences/risks for the interpretation of their results (and accordingly tone down the Discussion of such results).

A few more details should also be provided with respect to QTL mapping procedures. For instance, as far as I know, the EM algorithm for scanone and scantwo in R/qtl requires the computation of genotype probabilities, which can be in turn computed in various ways (at observed markers, or more frequently at intervals expressed in centiMorgans; with various mapping functions and so on). Similarly, the Authors should report explicitly whether the normal or the binary models have been used.

Minor issues

L13: I suggest "the dorsal portion" instead of "dorsal aspects"; also "to the chest and lower head"

L20-24: I suggest rephrasing the sentence so that it starts with “Pundamilia sp. “red-head” and Pundamilia pundamilia are”

L29: instead of “themselves”, I suggest “which”

L99: instead of “multiply”, either “multiple” or “repeated”

L295: “our” (or “the”) instead of “or”

References cited in the review

Feulner PGD, Schwarzer J, Haesler MP, Meier JI, Seehausen O. 2018. A Dense Linkage Map of Lake Victoria Cichlids Improved the Pundamilia Genome Assembly and Revealed a Major QTL for Sex-Determination. *G3: Genes | Genomes | Genetics* 8:2411-2420.

Flaxman SM, Wacholder AC, Feder JL, Nosil P. 2014. Theoretical models of the influence of genomic architecture on the dynamics of speciation. *Molecular Ecology* 23:4074-4088.

Fruciano C, Franchini P, Kovacova V, Elmer KR, Henning F, Meyer A. 2016. Genetic linkage of distinct adaptive traits in sympatrically speciating crater lake cichlid fish. *Nature communications* 7:12736.

Meier JI, Marques DA, Wagner CE, Excoffier L, Seehausen O. 2018. Genomics of Parallel Ecological Speciation in Lake Victoria Cichlids. *Molecular Biology and Evolution* 35:1489-1506.

Review form: Reviewer 2

Recommendation

Accept with minor revision (please list in comments)

Scientific importance: Is the manuscript an original and important contribution to its field?

Good

General interest: Is the paper of sufficient general interest?

Good

Quality of the paper: Is the overall quality of the paper suitable?

Good

Is the length of the paper justified?

Yes

Should the paper be seen by a specialist statistical reviewer?

Yes

Do you have any concerns about statistical analyses in this paper? If so, please specify them explicitly in your report.

No

It is a condition of publication that authors make their supporting data, code and materials available - either as supplementary material or hosted in an external repository. Please rate, if applicable, the supporting data on the following criteria.

Is it accessible?

Yes

Is it clear?

Yes

Is it adequate?

Yes

Do you have any ethical concerns with this paper?

No

Comments to the Author

RSPB-2019-2310

Title: Genetic architecture of a key reproductive isolation trait differs between sympatric and non-sympatric sister species of Lake Victoria cichlids

In this paper Feller et al. use two different cross lines (a naturally sympatric species pair and a non-sympatric one) to test for the potential role of genetic architecture in maintaining species boundaries. They find that the genetic architecture differs among the species pairs and suggest that indeed the difference may be a key trait in maintaining reproductive isolation.

I find the paper very well written and clear, the analyses and conclusion appear sound to me and I have only few relatively minor comments.

On a more general site I find some of the sentences in the introduction and discussion too long and complicated (e.g. lines 17ff, lines 323ff). A second more general issue I have is that the discussion lacks a bit of context beyond the work of the work group presenting this paper. I think it would be good to also discuss the knowledge on the other cichlid radiations to compare their results to. Finally, I think the methods need a few more details. Why some more is described in the appendix I wonder if at least things like restriction enzymes for RAD, the way of scoring phenotypes (manually, or program, how exactly?), etc. should be provided.

Besides I have a few more specific points which I list chronologically below:

Methods: Why were two different genotypers used for the different datasets? Are these comparable?

Line 158f: What was the final number of individuals? Not clear to me from this.

Line 168f: What is a ripple after addition? Is this a function of joinmap? Please specify.

Line 193: How was the random downsampling performed? Was a program/algorithm used?

Line 251ff: Is it possible to trace the QTLs to any specific genes by mapping respective loci to the existing genomes?

Figure 2. Would not red and grey always add to one? Then maybe just show the red bars?
Besides I think this is a nice and concise paper.

kind regards

Decision letter (RSPB-2019-2310.R0)

12-Nov-2019

Dear Miss Feller:

I am writing to inform you that your manuscript RSPB-2019-2310 entitled "Genetic architecture of a key reproductive isolation trait differs between sympatric and non-sympatric sister species of Lake Victoria cichlids" has, in its current form, been rejected for publication in Proceedings B.

This action has been taken on the advice of referees, who have recommended that substantial revisions are necessary. When deciding whether to reject and allow re-submission or request a revision, reject and allow resubmission provides more time to complete the revision. With this in mind we would be happy to consider a resubmission, provided the comments of the referees are fully addressed. However please note that this is not a provisional acceptance.

Sincerely,
Dr Daniel Costa
mailto: proceedingsb@royalsociety.org

Associate Editor
Board Member: 1
Comments to Author:

This study uses a comparative quantitative genetic approach to determine whether differences in the genetic architecture of male nuptial colouration explain the diversity and difference in the distribution patterns of East African cichlid species pairs. Using QTL mapping of these traits with a fairly extensive crossing design, they find evidence that the genetic architecture of a key reproductive isolation trait differs between sympatric and non-sympatric sister species of Lake Victoria cichlid.

The study received two reviews from experts in the field and I have read the paper myself. I agree with both Referees that the MS is potentially a really good fit for PRSB. The system is fascinating, the phenotyping was excellent and the study is an appropriate application of a top-down approach to test this important question. Nonetheless, both Referees raise important concerns that need to be addressed to confirm that the conclusions are supported by the data. Namely, there appear to have been different criteria and information used in the two crosses for QTL analysis. I agree with both Referees that these methods and data could have been better documented in the methods, while certain aspects will need revision to be convincing for readers that the data sets and corresponding QTL results between crosses are comparable and robust to

test these hypotheses. As it stands, there isn't quite enough information presented to gauge whether the differences in results between the two crosses reflects biological differences. My sense is the mapping resolution achieved in both crosses was sufficient to detect QTL (and supported by downsampling), but I agree with that this does not account for the potential impact and differences in quality/processing between the crosses that possibly had an impact on the analyses. Against the backdrop of biases that can arise in QTL detection and estimation of effect size, the Referee's suggestions are reasonable. The authors may want to take a look at Blankers et al. (PRSB 2019) where high confidence maps were constructed for the purpose of estimating QTL detection and reuse among closely related species pairs of crickets (with corresponding methods in QTL analyses to realistically reflect the conditions of their comparative QTL experiment).

Several other useful edits for clarity, to broaden the scope and in consideration of alternative hypotheses were also made by both Referees and should be taken into account when revising the MS. Because the combination of revisions includes changes to the text in addition to significant reanalyses, I am recommending that major revisions will be needed before resubmission.

Reviewer(s)' Comments to Author:

Referee: 1

Comments to the Author(s)

The manuscript "Genetic architecture of a key reproductive isolation trait differs between sympatric and non-sympatric sister species of Lake Victoria cichlids" by Feller and colleagues reports on two QTL mapping analyses on two crosses of closely related Lake Victoria cichlid fish.

The manuscript is overall interesting and well-written, and a great fit for Proceedings B so I think that this manuscript will eventually make a great addition to the journal.

For the time being, however, I have a few concerns which prevent me from suggesting acceptance. Addressing these concerns will require changes to the text and most likely partial reanalysis, but they are overall doable and should not undermine the main message of the manuscript. For this reason, I encourage the Authors to put in the necessary effort to improve this manuscript following my comments.

L38: The Authors should better qualify statements about strength of differentiation, particularly to the benefit of a broader audience which may not be familiar with the study system and/or typical divergence in sympatric settings.

Here, the Authors define the differentiation between "nyerei-like" and "pundamilia-like" as "surprisingly strong". However, figure 1b in Meier, et al. (2018) (which the Authors cite as source) reports for this species pair an F_{ST} of 0.053, which most people would find neither strong nor surprising.

I suggest that the Authors provide some information about how strong (F_{ST} or some other relevant results from the study they cite) and, possibly, tone down this statement.

A possibility is - in addition to providing some information about the strength of differentiation - also explaining to the reader why a certain level of differentiation is surprisingly strong in this system.

L49-51: This is correct, but another factor which could contribute to divergence in the presence of gene flow is (loosely defined) linkage of multiple traits, as the organization of genes in genomes will itself create non-random combinations of alleles across multiple loci, and this clearly will be even more fitting in the case of multiple traits being subject to selection and mapping to the same region.

While it is somewhat debatable to which level the different traits mapped in this study represent distinct traits, I think that the explanation above should be considered and explicitly acknowledged in the Introduction and Discussion. This is because of the substantial level of overlap in QTL regions the Authors find in their study (e.g., Figure 3c).

Relevant work which could be included in such discussion include Fruciano, et al. (2016) for an empirical case in a sympatric system.

Flaxman, et al. (2014) could also be cited for simulation/theory on the general idea that linkage/organization in genomes will favour divergence (which is reflected in the next sentence at L55). The reason why I think Flaxman et al's work is relevant here is that, while the Authors of this manuscript focus on maintenance of phenotypic variation in the face of gene flow (L55-57), the very young age of this system makes it possible that we are now observing a "snapshot" of divergence between two forming species which in the future would go through a quick transition "congealing" phase as per Flaxman et al's simulations, possibly aided by linkage of multiple traits.

Sequence processing, genotyping and map construction - I find that the use of different criteria for the two crosses is neither adequately documented nor justified.

For instance, at lines 154-155, the Authors simply state that the procedure for the red-chest x blue cross was very similar to the one for the other cross, but with a different software for genotyping. However, checking the original paper by Feulner, et al. (2018) the procedure appears substantially different compared to the one in Appendix S2. For instance, Feulner, et al. (2018) use an initial depth threshold of 3 (which is actually quite low) while in Appendix S2 genotypes with a depth <10 are set to missing.

This creates a situation in which the level of reliability of the two datasets/crosses is different (it is hard to evaluate by how much, as the sequential procedures used are different). Then, the radically different number of markers in the maps for the two crosses (368 vs 2052) are probably not only a function of the different level of divergence between parental species, but also probably the inclusion of lower-quality SNPs in the red-chest x blue cross.

The same applies to map construction. While map construction procedure is similar between the two studies, the thresholds and criteria used here are different from the ones in Feulner, et al. (2018).

All of this creates a substantial risk that differences in results between the two crosses are due to difference in statistical power, rather than to real biological differences. I do appreciate and commend the Authors for the downsampling solution they have included (L188-195). However, this solution still does not account for differences in quality/processing between crosses.

First of all, sufficient information on the red-chest x blue cross should be provided in this manuscript/supplementary to avoid that the reader has to check a different paper.

Second, my strongly preferred suggestion is that the analyses are repeated using the same criteria/pipeline for the red-chest x blue cross to ensure a fair comparison between the two crosses.

As a much less preferred option, the Authors should to the very least be very clear to the reader about what the differences between the two crosses are, and what are the consequences/risks for the interpretation of their results (and accordingly tone down the Discussion of such results).

A few more details should also be provided with respect to QTL mapping procedures. For instance, as far as I know, the EM algorithm for scanone and scantwo in R/qtl requires the computation of genotype probabilities, which can be in turn computed in various ways (at observed markers, or more frequently at intervals expressed in centiMorgans; with various mapping functions and so on). Similarly, the Authors should report explicitly whether the normal or the binary models have been used.

Minor issues

L13: I suggest "the dorsal portion" instead of "dorsal aspects"; also "to the chest and lower head"

L20-24: I suggest rephrasing the sentence so that it starts with "Pundamilia sp. "red-head" and Pundamilia pundamilia are"

L29: instead of “themselves”, I suggest “which”
 L99: instead of “multiply”, either “multiple” or “repeated”
 L295: “our” (or “the”) instead of “or”

References cited in the review

Feulner PGD, Schwarzer J, Haesler MP, Meier JI, Seehausen O. 2018. A Dense Linkage Map of Lake Victoria Cichlids Improved the Pundamilia Genome Assembly and Revealed a Major QTL for Sex-Determination. *G3: Genes | Genomes | Genetics* 8:2411-2420.
 Flaxman SM, Wacholder AC, Feder JL, Nosil P. 2014. Theoretical models of the influence of genomic architecture on the dynamics of speciation. *Molecular Ecology* 23:4074-4088.
 Fruciano C, Franchini P, Kovacova V, Elmer KR, Henning F, Meyer A. 2016. Genetic linkage of distinct adaptive traits in sympatrically speciating crater lake cichlid fish. *Nature communications* 7:12736.
 Meier JI, Marques DA, Wagner CE, Excoffier L, Seehausen O. 2018. Genomics of Parallel Ecological Speciation in Lake Victoria Cichlids. *Molecular Biology and Evolution* 35:1489-1506.

Referee: 2

Comments to the Author(s)
 RSPB-2019-2310

Title: Genetic architecture of a key reproductive isolation trait differs between sympatric and non-sympatric sister species of Lake Victoria cichlids

In this paper Feller et al. use two different cross lines (a naturally sympatric species pair and a non-sympatric one) to test for the potential role of genetic architecture in maintaining species boundaries. They find that the genetic architecture differs among the species pairs and suggest that indeed the difference may be a key trait in maintaining reproductive isolation.

I find the paper very well written and clear, the analyses and conclusion appear sound to me and I have only few relatively minor comments.

On a more general site I find some of the sentences in the introduction and discussion too long and complicated (e.g. lines 17ff, lines 323ff). A second more general issue I have is that the discussion lacks a bit of context beyond the work of the work group presenting this paper. I think it would be good to also discuss the knowledge on the other cichlid radiations to compare their results to. Finally, I think the methods need a few more details. Why some more is described in the appendix I wonder if at least things like restriction enzymes for RAD, the way of scoring phenotypes (manually, or program, how exactly?), etc. should be provided.

Besides I have a few more specific points which I list chronologically below:

Methods: Why were two different genotypers used for the different datasets? Are these comparable?

Line 158f: What was the final number of individuals? Not clear to me from this.

Line 168f: What is a ripple after addition? Is this a function of joinmap? Please specify.

Line 193: How was the random downsampling performed? Was a program/algorithm used?

Line 251ff: Is it possible to trace the QTLs to any specific genes by mapping respective loci to the existing genomes?

Figure 2. Would not red and grey always add to one? Then maybe just show the red bars?
Besides I think this is a nice and concise paper.

kind regards

Author's Response to Decision Letter for (RSPB-2019-2310.R0)

See Appendix A.

RSPB-2020-0270.R0

Review form: Reviewer 1

Recommendation

Accept as is

Scientific importance: Is the manuscript an original and important contribution to its field?

Good

General interest: Is the paper of sufficient general interest?

Good

Quality of the paper: Is the overall quality of the paper suitable?

Good

Is the length of the paper justified?

Yes

Should the paper be seen by a specialist statistical reviewer?

No

Do you have any concerns about statistical analyses in this paper? If so, please specify them explicitly in your report.

No

It is a condition of publication that authors make their supporting data, code and materials available - either as supplementary material or hosted in an external repository. Please rate, if applicable, the supporting data on the following criteria.

Is it accessible?

No

Is it clear?

Yes

Is it adequate?

N/A

Do you have any ethical concerns with this paper?

No

Comments to the Author

I commend the Authors for the work they have done on the manuscript and how diligently and professionally they followed my suggestions. I am also glad that - as I had predicted - the main results are robust to the new analyses.

For these reasons, I believe that the manuscript can now be accepted.

I have a few, very minor, comments on things that - as far as I'm concerned - could be even fixed at the proof stage.

- The Authors correctly document now the computation of genotype probabilities, with the step size. They should - however - include the unit. This will depend on the units of the map, but typically is in centiMorgans
- L223 "respective" can be removed
- L281 "for any"
- L374 I suggest "single pleiotropic"

Decision letter (RSPB-2020-0270.R0)

09-Mar-2020

Dear Miss Feller

I am pleased to inform you that your manuscript RSPB-2020-0270 entitled "Genetic architecture of a key reproductive isolation trait differs between sympatric and non-sympatric sister species of Lake Victoria cichlids" has been accepted for publication in Proceedings B.

The referee(s) have recommended publication, but also suggest some minor revisions to your manuscript. Therefore, I invite you to respond to the referee(s)' comments and revise your manuscript. Because the schedule for publication is very tight, it is a condition of publication that you submit the revised version of your manuscript within 7 days. If you do not think you will be able to meet this date please let us know.

- 1) A text file of the manuscript (doc, txt, rtf or tex), including the references, tables (including

captions) and figure captions. Please remove any tracked changes from the text before submission. PDF files are not an accepted format for the "Main Document".

2) A separate electronic file of each figure (tiff, EPS or print-quality PDF preferred). The format should be produced directly from original creation package, or original software format. PowerPoint files are not accepted.

3) Electronic supplementary material: this should be contained in a separate file and where possible, all ESM should be combined into a single file. All supplementary materials accompanying an accepted article will be treated as in their final form. They will be published alongside the paper on the journal website and posted on the online figshare repository. Files on figshare will be made available approximately one week before the accompanying article so that the supplementary material can be attributed a unique DOI.

Sincerely,

Dr Daniel Costa
mailto: proceedingsb@royalsociety.org

Associate Editor
Board Member
Comments to Author:

This authors have thoroughly addressed the issues brought up in the previous review. The comparative quantitative genetic approach is now consistent and robust, and the conclusions about genetic architecture better supported by the data. I recommend Accept following the very minor revisions caught by the Referee.

Reviewer(s)' Comments to Author:

Referee: 1

Comments to the Author(s).

I commend the Authors for the work they have done on the manuscript and how diligently and professionally they followed my suggestions. I am also glad that - as I had predicted - the main results are robust to the new analyses.

For these reasons, I believe that the manuscript can now be accepted.

I have a few, very minor, comments on things that - as far as I'm concerned - could be even fixed at the proof stage.

- The Authors correctly document now the computation of genotype probabilities, with the step size. They should - however - include the unit. This will depend on the units of the map, but typically is in centiMorgans
- L223 "respective" can be removed
- L281 "for any"
- L374 I suggest "single pleiotropic"

Author's Response to Decision Letter for (RSPB-2020-0270.R0)

See Appendix B.

Decision letter (RSPB-2020-0270.R1)

17-Mar-2020

Dear Miss Feller

I am pleased to inform you that your manuscript entitled "Genetic architecture of a key reproductive isolation trait differs between sympatric and non-sympatric sister species of Lake Victoria cichlids" has been accepted for publication in Proceedings B.

Open Access

Paper charges

Sincerely,

Appendix A

Response to referees

We would like to thank both referees and the editor for their positive and constructive feedback. We have addressed the issues raised in our point-by-point responses below. Line numbers correspond to the revised version of the manuscript, and as requested, we have also attached a track-change version of the main text.

Referee 1

The manuscript “Genetic architecture of a key reproductive isolation trait differs between sympatric and non-sympatric sister species of Lake Victoria cichlids” by Feller and colleagues reports on two QTL mapping analyses on two crosses of closely related Lake Victoria cichlid fish. The manuscript is overall interesting and well-written, and a great fit for Proceedings B so I think that this manuscript will eventually make a great addition to the journal. For the time being, however, I have a few concerns which prevent me from suggesting acceptance. Addressing these concerns will require changes to the text and most likely partial reanalysis, but they are overall doable and should not undermine the main message of the manuscript. For this reason, I encourage the Authors to put in the necessary effort to improve this manuscript following my comments.

Comment R1.1: L38: The Authors should better qualify statements about strength of differentiation, particularly to the benefit of a broader audience which may not be familiar with the study system and/or typical divergence in sympatric settings. Here, the Authors define the differentiation between “nyerei-like” and “pundamilia-like” as “surprisingly strong”. However, figure 1b in Meier, et al. (2018) (which the Authors cite as source) reports for this species pair an FST of 0.053, which most people would find neither strong nor surprising. I suggest that the Authors provide some information about how strong (FST or some other relevant results from the study they cite) and, possibly, tone down this statement. A possibility is – in addition to providing some information about the strength of differentiation – also explaining to the reader why a certain level of differentiation is surprisingly strong in this system.

An FST value of 0.053 between *Pundamilia* sp. “nyerei-like” and *Pundamilia* sp. “pundamilia-like” is surprisingly high considering the age of this fully sympatric species pair is estimated to be less than 200 generations (Meier *et al.*, 2017) and they have likely evolved in full sympatry (Meier *et al.*, 2017, 2018). Furthermore, the differentiation between these species is characterized by hundreds of highly differentiated genomic regions (Meier *et al.*, 2018). We have added these explanations on lines 69-72.

Comment R1.2: L49-51: This is correct, but another factor which could contribute to divergence in the presence of gene flow is (loosely defined) linkage of multiple traits, as the organization of genes in genomes will itself create non-random combinations of alleles across multiple loci, and this clearly will be even more fitting in the case of multiple traits being subject to selection and mapping to the same region. While it is somewhat debatable to which level the different traits mapped in this study represent distinct traits, I think that the explanation above should be considered and explicitly acknowledged in the Introduction and Discussion. This is because of the substantial level of overlap in QTL regions the Authors find in their study (e.g., Figure 3c). Relevant work which could be included in such discussion include Fruciano, et al. (2016) for an empirical case in a sympatric system. Flaxman, et al. (2014) could also be cited for simulation/theory on the general idea that linkage/organization in genomes will favour divergence (which is reflected in the next sentence at L55). The reason why I think Flaxman et al’s work is relevant here is that, while the Authors of this manuscript focus on maintenance of phenotypic variation in the face of gene flow (L55-57), the very young age of this system makes it possible that we are now observing a “snapshot” of divergence between two forming species which in the future would go through a quick transition “congealing” phase as per Flaxman et al’s simulations, possibly aided by linkage of multiple traits.

We agree that linkage (or pleiotropy) is another factor that could facilitate species divergence and persistence in sympatry with gene flow, which we had not yet explicitly addressed in our discussion. We have added a sentence concerning linkage in the

Introduction (lines 91-92), and more explicitly discuss this scenario as suggested in the Discussion (lines 378-385). Also see lines 97-98, 112-117, 408, and the abstract, for minor edits to include linkage and pleiotropy in the overall discussion.

Comment R1.3: Sequence processing, genotyping and map construction – I find that the use of different criteria for the two crosses is neither adequately documented nor justified. For instance, at lines 154-155, the Authors simply state that the procedure for the red-chest x blue cross was very similar to the one for the other cross, but with a different software for genotyping. However, checking the original paper by Feulner, et al. (2018) the procedure appears substantially different compared to the one in Appendix S2. For instance, Feulner, et al. (2018) use an initial depth threshold of 3 (which is actually quite low) while in Appendix S2 genotypes with a depth <10 are set to missing. This creates a situation in which the level of reliability of the two datasets/crosses is different (it is hard to evaluate by how much, as the sequential procedures used are different). Then, the radically different number of markers in the maps for the two crosses (368 vs 2052) are probably not only a function of the different level of divergence between parental species, but also probably the inclusion of lower-quality SNPs in the red-chest x blue cross. The same applies to map construction. While map construction procedure is similar between the two studies, the thresholds and criteria used here are different from the ones in Feulner, et al. (2018). All of this creates a substantial risk that differences in results between the two crosses are due to difference in statistical power, rather than to real biological differences. I do appreciate and commend the Authors for the downsampling solution they have included (L188-195). However, this solution still does not account for differences in quality/processing between crosses. First of all, sufficient information on the red-chest x blue cross should be provided in this manuscript/supplementary to avoid that the reader has to check a different paper. Second, my strongly preferred suggestion is that the analyses are repeated using the same criteria/pipeline for the red-chest x blue cross to ensure a fair comparison between the two crosses. As a much less preferred option, the Authors should to the very least be very clear to the reader about what the differences between the two crosses are, and what are the consequences/risks for the interpretation of their results (and accordingly tone down the Discussion of such results).

We thank Referee 1 for this detailed feedback and the constructive suggestions. Following the reviewer's recommendations (strongly preferred option), we have repeated all analyses in the red-chest x blue cross using the exact same pipeline with the same criteria as used and already described in the first version of the manuscript for the red-dorsum x blue cross, from raw sequences to linkage map construction and QTL mapping. The Methods (including Appendix) and Results, including Figures 2 & 3 and the Tables (in the Supplementary) have been revised accordingly. The overall result has not changed and our conclusions remain robust.

A more detailed report below:

The number of markers that appeared as homozygous fixed (and heterozygous in the F1) in the red-chest x blue cross was higher than in (Feulner *et al.*, 2018) due to the application of the same allelic balance filter used in the red-dorsum x blue cross (lines 190-192, and see Appendix S2). However, most of these additional markers and more were removed again in the stringent filtering procedure in linkage map construction. The final linkage map contains 1,198 markers (lines 246-248). Hence, the large difference in marker numbers persists even when filtering with the same criteria, indicating this is a real difference between the two crosses.

Due to the different filters applied, the number of individuals in the red-chest x blue cross has dropped slightly from 132 individuals with both phenotype and genotype data to 125, resulting in minimal changes in the phenotype distributions (revised Figure 2), and one less mappable trait (red on the rear part of the dorsal fin, lines 264-267).

We repeated the downsampling procedure for the red-dorsum x blue cross with this new somewhat smaller number of individuals. This re-analysis still recovered most of

the significant QTLs that we had obtained in this cross with the complete data set (82%; lines 293-296).

The new QTL mapping results (lines 297-309, revised Tables S1, S2, Figure 3b) for the red-chest x blue cross consistently fail to reveal any significant QTLs. There are some changes in which traits reached marginal significance and which did not: In the original QTL analysis (as reported in the original submission), red on the nose, cheek (only with family as covariate), ventrum and upper lip (both only in some mapping rounds with the subsampled maps) reached marginal significance, and one trait (nose) appeared in the two-dimensional QTL scans. In the new QTL analysis, red on the throat (only with family as additive covariate), and head (only in some mapping rounds with the subsampled maps) reached marginal significance, and three traits (cheek, throat and yellow flanks) appeared in the two-dimensional QTL scans. Fluctuations in which traits reach marginal significance and which do not may not be surprising considering the P-values and significance threshold are obtained by permutation. These marginally significant QTLs should probably not be overinterpreted. Taken at face value, they are consistent with the interpretation that the traits that we mapped are indeed heritable also in the red-chest x blue cross (as indeed we know from many years of breeding these species in the lab) but are determined by genes with small effects without evidence for major effect genes in this cross.

Comment R1.4: A few more details should also be provided with respect to QTL mapping procedures. For instance, as far as I know, the EM algorithm for scanone and scantwo in R/qtl requires the computation of genotype probabilities, which can be in turn computed in various ways (at observed markers, or more frequently at intervals expressed in centiMorgans; with various mapping functions and so on). Similarly, the Authors should report explicitly whether the normal or the binary models have been used.

We have added these details in the Methods section on lines 211-214 (*“Conditional genotype probabilities were calculated using the calc.genoprob function with a fixed stepsize of 1 (respectively 3 for scantwo), an assumed genotyping error rate of 0.05, and the Kosambi map function.”*) and on lines 215-217 (*“For the red/yellow traits we used the binary model, for the hybrid index we used the normal model.”*) We have also added more details on calculating PVE on lines 223-226.

Comment R1.5: Minor issues

L13: I suggest “the dorsal portion” instead of “dorsal aspects”; also “to the chest and lower head”

L20-24: I suggest rephrasing the sentence so that it starts with “Pundamilia sp. “red-head” and Pundamilia pundamilia are”

L29: instead of “themselves”, I suggest “which”

L99: instead of “multiply”, either “multiple” or “repeated”

L295: “our” (or “the”) instead of “or”

We have implemented these suggestions and fixed the typo, and thank Referee 1 for pointing them out to us.

Referee 2

In this paper Feller et al. use two different cross lines (a naturally sympatric species pair and a non-sympatric one) to test for the potential role of genetic architecture in maintaining species boundaries. They find that the genetic architecture differs among the species pairs and suggest that indeed the difference may be a key trait in maintaining reproductive isolation. I find the paper very well written and clear, the analyses and conclusion appear sound to me and I have only few relatively minor comments.

Comment R2.1: On a more general site I find some of the sentences in the introduction and discussion too long and complicated (e.g. lines 17ff, lines 323ff).

We have broken up these long sentences into smaller pieces, e.g. lines 48ff and 66ff in the Introduction, and lines 312ff in the Discussion.

Comment R2.2: A second more general issue I have is that the discussion lacks a bit of context beyond the work of the work group presenting this paper. I think it would be good to also discuss the knowledge on the other cichlid radiations to compare their results to.

We (with space limitation in mind) have reorganized the Discussion such that the comparison to other systems should be more apparent, and (also in line with Comment R1.2.) added another cichlid example (e.g. lines 383-385).

Comment R2.3: Finally, I think the methods need a few more details. Why some more is described in the appendix I wonder if at least things like restriction enzymes for RAD, the way of scoring phenotypes (manually, or program, how exactly?), etc. should be provided. Besides I have a few more specific points which I list chronologically below:

We have now provided details concerning restriction enzyme on lines line 141 (“...using *SbfI* as restriction enzyme...”) and concerning the scoring of phenotypes, which was done visually (i.e. manually) on line 172.

Comment R2.4a: Methods: Why were two different genotypers used for the different datasets? Are these comparable?

We agree that it is not entirely clear how comparable the two genotypers are, and we have now used the exact same pipeline including the same genotyper with the same settings/criteria for both datasets, as elaborated in more detail above (see response to Comment R1.3).

Comment R2.4b: Line 158f: What was the final number of individuals? Not clear to me from this.

We have now added this information on lines 197-199: “*The linkage maps were generated from 216 F2 individuals (173 males, 43 females) in the red-dorsum x blue cross and 171 F2 individuals (115 males, 56 females) in the red-chest x blue cross.*”

Comment R2.4c: Line 168f: What is a ripple after addition? Is this a function of joinmap? Please specify.

Yes, this is a function of JoinMap. In regression mapping, the loci are added to the map one by one. By performing a ripple when a locus is added, all permutations within a moving window of three adjacent markers are considered and their goodness-of-fit calculated to determine the best order to continue with the map (van Ooijen, 2006). We did not add any additional details of this function, as it is a default function when creating linkage maps using JoinMap.

Comment R2.4d: Line 193: How was the random downsampling performed? Was a program/algorithm used?

We have added details on downsampling in the Methods section on lines 230-238: “(...) we repeated the single QTL analyses for the red-dorsum x blue cross after randomly downsampling to 125 (of 174) individuals using the sample function in R to randomly pick individuals and then subsetting the genotype-phenotype file to these individuals. For the red-chest x blue cross we repeated the single QTL analyses after randomly downsampling markers on the linkage map to match the number of markers on each linkage group to those in our sparser red-dorsum x blue map, again using the sample function in R to randomly pick markers within each linkage group and then subsetting them to these sampled markers. The respective procedure was repeated five times in each cross.” In line with this we have also added the missing “R” reference.

Comment R2.4e: Line 251ff: Is it possible to trace the QTLs to any specific genes by mapping respective loci to the existing genomes?

The intervals of significant QTLs in the red-dorsum x blue cross are so large that they each contain dozens, and in most cases even hundreds of known genes. Our first screening of the genes closest to the QTL peaks did not reveal any obvious candidates. To follow up on these regions is thus out of the scope of this paper and will be a topic of future work. We added two sentences on this in the Discussion on lines 370-372.

Comment R2.4f: Figure 2. Would not red and grey always add to one? Then maybe just show the red bars?

We thank Referee 2 for this suggestion. However, this is complicated by the fact that the number of individuals with scores is not the same for each trait, as some individuals could not be scored unambiguously for some traits and were assigned an NA for such traits (see main text lines 162-164), so in this sense it does not always add up to one. In addition to that, we think it is easier to see the proportion of presence (red) vs absence (grey) scores by having both bars next to each other. For better visibility we have increased the gaps between paired bars.

Associate Editor

This study uses a comparative quantitative genetic approach to determine whether differences in the genetic architecture of male nuptial colouration explain the diversity and difference in the distribution patterns of East African cichlid species pairs. Using QTL mapping of these traits with a fairly extensive crossing design, they find evidence that the genetic architecture of a key reproductive isolation trait differs between sympatric and non-sympatric sister species of Lake Victoria cichlid. The study received two reviews from experts in the field and I have read the paper myself. I agree with both Referees that the MS is potentially a really good fit for PRSB. The system is fascinating, the phenotyping was excellent and the study is an appropriate application of a top-down approach to test this important question. Nonetheless, both Referees raise important concerns that need to be addressed to confirm that the conclusions are supported by the data.

Comment E.1: Namely, there appear to have been different criteria and information used in the two crosses for QTL analysis. I agree with both Referees that these methods and data could have been better documented in the methods, while certain aspects will need revision to be convincing for readers that the data sets and corresponding QTL results between crosses are comparable and robust to test these hypotheses. As it stands, there isn't quite enough information presented to gauge whether the differences in results between the two crosses reflects biological differences. My sense is the mapping resolution achieved in both crosses was sufficient to detect QTL (and supported by downsampling), but I agree with that this does not account for the potential impact and differences in quality/processing between the crosses that possibly had an impact on the analyses. Against the backdrop of biases that can arise in QTL detection and estimation of effect size, the Referee's suggestions are reasonable. The authors may want to take a look at Blankers et al. (PRSB 2019) where high confidence maps were constructed for the purpose of estimating QTL detection and reuse among closely related species pairs of crickets (with corresponding methods in QTL analyses to realistically reflect the conditions of their comparative QTL experiment).

As described in detail under Comment R1.3, we have now repeated all analyses in both crosses with the same pipeline and criteria. Our overall result and conclusions remain the same. This shows that the results are robust to different versions of genotype calling and to different linkage map versions (including downsampled maps), and thus we are confident that the difference in the genetic architecture of a key reproductive isolation trait between the two crosses is real.

Comment E.2: Several other useful edits for clarity, to broaden the scope and in consideration of alternative hypotheses were also made by both Referees and should be taken into account when revising the MS. Because the combination of revisions includes changes to the text in addition to significant reanalyses, I am recommending that major revisions will be needed before resubmission.

We have implemented these edits as indicated point by point above.

References

- Feulner, P.G.D., Schwarzer, J., Haesler, M.P., Meier, J.I. & Seehausen, O. 2018. Data from: A dense linkage map of Lake Victoria cichlids improved the *Pundamilia* genome assembly and revealed a major QTL for sex-determination. Dryad Digital Repository.
- Meier, J.I., Marques, D.A., Wagner, C.E., Excoffier, L. & Seehausen, O. 2018. Genomics of parallel ecological speciation in Lake Victoria cichlids. *Mol. Biol. Evol.* **35**: 1489–1506.
- Meier, J.I., Sousa, V.C., Marques, D.A., Selz, O.M., Wagner, C.E., Excoffier, L., *et al.* 2017. Demographic modelling with whole-genome data reveals parallel origin of similar *Pundamilia* cichlid species after hybridization. *Mol. Ecol.* **26**: 123–141.
- van Ooijen, J.W. 2006. JoinMap®4, software for the calculation of genetic linkage maps in experimental populations.

Appendix B

Response to referees

Associate Editor

This authors have thoroughly addressed the issues brought up in the previous review. The comparative quantitative genetic approach is now consistent and robust, and the conclusions about genetic architecture better supported by the data. I recommend Accept following the very minor revisions caught by the Referee.

Referee 1

I commend the Authors for the work they have done on the manuscript and how diligently and professionally they followed my suggestions. I am also glad that - as I had predicted - the main results are robust to the new analyses. For these reasons, I believe that the manuscript can now be accepted. I have a few, very minor, comments on things that - as far as I'm concerned - could be even fixed at the proof stage.

- The Authors correctly document now the computation of genotype probabilities, with the step size. They should - however - include the unit. This will depend on the units of the map, but typically is in centiMorgans

- L223 "respective" can be removed

- L281 "for any"

- L374 I suggest "single pleiotropic"

We would again like to thank the referees and the editor for their positive and constructive feedback. We have implemented the three text corrections suggested by referee 1, and we have now included the step size unit (cM) in the documentation of the computation of genotype probabilities. A "tracked changes" copy of the revised manuscript is attached below as requested.